# A Simplified Mathematical Model for the Analysis of Varying Compliance Vibrations of a Rolling Bearing

**Radoslav Tomović** 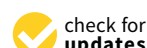

Mechanical Engineering Faculty, University of Montenegro, 81000 Podgorica, Montenegro; radoslav@ucg.ac.me

**Abstract:** In this paper, a simplified approach in the analysis of the varying compliance vibrations of a rolling bearing is presented. This approach analyses the generation of vibrations in relation to two boundary positions of the inner ring support on an even and an odd number of the rolling element of a bearing. In this paper, a mathematical model for the calculation of amplitude and frequency of vibrations of a rigid rotor in a rolling bearing is presented. The model is characterized by a big simplicity which makes it very convenient for a practical application. Based on the presented mathematical model a parametric analysis of the influence of the internal radial clearance, external radial load and the total number of rolling elements on the varying compliance vibrations of rolling bearing was conducted. These parameters are the most influential factors for generating varying compliance vibrations. The results of the parametric analysis demonstrate that with the proper choice of the size of the internal radial clearance and external radial load, the level of the varying compliance vibrations in a rolling bearing can be theoretically reduced to zero. This result opposes the opinion that varying compliance vibrations of rolling bearing cannot be avoided, even for geometrically ideally produced bearing.

**Keywords:** rolling bearing; rigid rotor; internal radial clearance; number of rolling elements; vibration; varying compliance vibration; ball passage frequency; load

---

## 1. Introduction

The study of the motion of the rotational machine, i.e., rotor dynamics, is one of the most important areas in the engineering practice. The research of rotor dynamics finds its use in a broad spectrum of applications, from the big machines for energy production to those small ones found in the medical equipment. The mathematical models of a rotor and simulating the response to the activity of disturbing forces means a big aid in the developmental process of new structures and removal of problem related to already existing structures. The characteristics of a bearing where rotors are supported have a big influence on the rotor dynamic. The rolling bearings are the most used elements for the support of rotational machines. One of the most important problems related to rotational machines is the reduction of vibration and the increase of the accuracy of the spin of rotors supported on the rolling bearings, especially for the rotors operating with high rotational speeds. The rotation precision is often conditioned with the characteristics of vibration that are generated in the rolling bearings. Because of that, the problems related to the generation of vibrations in the rolling bearings should be treated with particular attention.

The main causes for vibration generation in rolling bearings are as follows [1]:

- The specific construction and the specific way of functioning of rolling bearing (varying compliance (VC) vibrations-structural vibrations),
- Micro and macro geometry errors of bearing elements (vibrations with technological origin),
- Damage of bearing elements (vibrations due to damage of bearings elements).

The construction of bearing affects the vibration generation in all the three mentioned cases. However, only in the (VC) vibrations, the construction of the bearing is the main cause of vibrations. Some hold the opinion that the VC vibrations in bearings cannot be avoided, even in ideally made rolling bearings and even in the absence of an external load [2,3]. In the other two cases, the vibrations occur because of various imperfections or damage which emerge during the fabrication or exploitation of bearings. With the correct production methods, maintenance, and exploitation, these vibrations can be reduced, and can even be avoided.

The first systematic researches varying compliance (VC) vibrations were done by Sunnersjö [4]. Sunnersjö studied the model of bearing considering the Hertzian contact and the clearance nonlinearities. The contact interaction between rolling elements and races is simplified as non-linear springs. Sunnersjö concluded that VC vibrations occur on the ball passage frequency (BPF) and its harmonics. Rahnejat and Gohar [5] later showed that even in the presence of elastohydrodynamic lubricating film between balls and races, a peak at the BPF appears in the spectrum.

The model proposed by Sunnersjö is a 2DOF model for bearing-rotor systems to study the varying compliance. This model and its modifications were later widely used in research by a large number of authors [5–15]. Mostly these are 2DOF models, however, 4DOF and 5DOF models were also used lately.

So, for example, Lynagh et al. [6] developed a two-dimensional model of the dynamic behavior of a rigid rotor in rolling bearings by observing the balance equation of bearing assembly as a system of elastically connected masses. Using this model, they have studied the problem of shaft rotation accuracy of highly precise machine tools at high rotation speeds. They also concluded that vibrations at BPF were dominant in the vibration spectrum. This observation was also confirmed by the experiment.

Tiwari M., Gupta K., and Prakash O. in [7] use the 2DOF model to study the effect of radial internal clearance on the dynamics behavior of a rigid rotor. They concluded that internal clearance is the biggest cause of unstable behavior of a rigid rotor and that subharmonic and chaotic behavior is strongly dependent on the size of the radial clearance. The study of the effect of clearance on a bearing's dynamic performance, quality, and operating life has gained a lot of attention lately because of the development of high-speed rotors [8]. Clearance non-linearity is different from most of the other non-linearities because it cannot be approximated by a mathematical series [7]. Upadhyay et al. [9] and Zhang et al. [10] also used a modified 2DOF model to study the nonlinear phenomena and chaotic behavior of a rigid rotor in a roller bearing.

The first model with 5DOF was developed by Aini et al. [11] and they applied it to the machine tool spindles. They have represented the bearings by nonlinear springs included elastohydrodynamic effects. This investigation was later refined by Li et al. [12], proposing a general dynamic model of the ball bearing-rotor system. They used the finite element method to model the contact stiffness. A little later, Wang et al. [13] propose their 5DOF dynamic model of the rotor system supported by angular contact ball bearings.

Five-degree-of-freedom models are used by many researchers to study the influence of various factors on the vibration generation of bearings. Thus, for example, Zhang et al. [14] study the effect of preload and varying contact angle on bearing vibration. Zhenhuan and Liqin [15] study the influence of rings misalignment on the dynamic characteristics of cylindrical roller bearings, and Yang et al. [16] influence of different ball number and rotor eccentricity on VC vibrations.

However, most of the above studies mainly merely verify the influence of the different factors on vibrations in the rotor-bearing system. Many studies only state that VC vibrations exist in the rotor-bearing system and that they occur on the BPF and that certain factors have an influence on the VC vibrations amplitude. Parametric analysis is very rarely performed.

The reason for this lies primarily in the great complexity of these models. All of these models are distinguished by their expressive non-linearity. Most of the mentioned models are encompass systems of differential equations, which describe the complex movement of the rigid rotor in rolling bearing, including the problem of nonlinear elastohydrodynamic contact between rolling elements and races. For solving systems of differential equations, most authors use the Newton–Raphson method, while

for obtaining a vibration spectrum the fast Fourier's transformation (FFT) is commonly used. These models are extremely difficult for solving and they require a large number of complicated mathematical operations. Because of that, any parametric analysis would take a very long time.

In this connection, the model of Kryuchkov and Smirnov is particularly interesting because of its simplicity. Kryuchkov [17] investigated the VC vibrations of a rigid rotor in an ideal rolling bearing with an internal radial clearance and found that these vibrations can be represented by the equation of the absolute sinusoid. The frequency of these vibrations is equal to the ball passage frequency (BPF). Smirnov [18] further developed the research of Kryuchkov.

Using the research of Kryuchkov, in this paper, a new mathematical model for vibration response prediction of a rigid rotor in rolling bearing has been presented. The simple mathematical record makes the developed model very convenient for practical usage. The model is based on a new approach in the analysis of rolling bearing which is presented in the papers [19–22]. For the time being, the model considers only the VC vibrations of rolling bearing and it refers to an ideally made bearing with the radial contact and internal radial clearance.

Based on the presented model a parametric analysis of mutual influence between internal radial clearance and external radial load on the VC vibrations of rolling bearing is conducted. These two parameters are the most influential factors for generating VC vibrations for a given type of rolling bearing [23].

However, to understand and describe this model in a reasonable manner, as the first step, it is necessary to explain the basic mechanisms for the generation of the VC vibrations in rolling bearings.

## 2. Mechanisms for the Generation of Varying Compliance (VC) Vibrations in Rolling Bearings

Varying compliance (VC) vibrations in rolling bearings are a consequence of the bearing discrete structure and specific mode of operation. In literature, they are often called the primary bearing induced vibration [5–7] or structural vibrations [24,25]. In principle, VC vibrations appear based on two causes:

- Due to the periodic oscillations of the center of the bearing inner ring which resulting from the change in the angular position of the rolling elements. The internal radial clearance is the primary cause of these vibrations. These vibrations were analysed by Harris [1], Sunnersjö [4], Kryuchkov [17], Tomović et al. [20], and Datta and Farhang [24,25].
- Due to the periodic change of the rolling bearing stiffness. These vibrations are known as variable compliance vibrations. They are discussed by Harris [1], Sunnersjö [4], Kryuchkov [17], Rahnejat and Gohar [5], Lynagh et al. [6], Tiwari M., Gupta K., and Prakash O. [7], Upadhyay et al. [9], Zhang et al. [10], Li et al. [12], Wang et al. [13], Yang et al. [15], etc.

The rotor vibrations that are the direct consequence of an internal radial clearance influence and the discrete structure of a rolling bearing will emerge in all bearings with the internal radial clearance, both under the load and without it. The vibrations occurred due to a time-varying stiffness that will emerge only in the loaded rolling bearings [19].

### 2.1. Vibrations Caused by the Influence of Internal Radial Clearance

During the operation of the rolling bearings, in relation to the direction of the external load action, the set of rolling elements continuously oscillates between the two end positions of the support shown in Figure 1. This phenomenon is described in detail in [19–22], where the following terms were adopted for these boundary positions:

- Boundary case of inner ring support on an odd number of rolling elements (BSO) (Figure 1a),
- Boundary case of inner ring support on an even number of rolling elements (BSE) (Figure 1b).

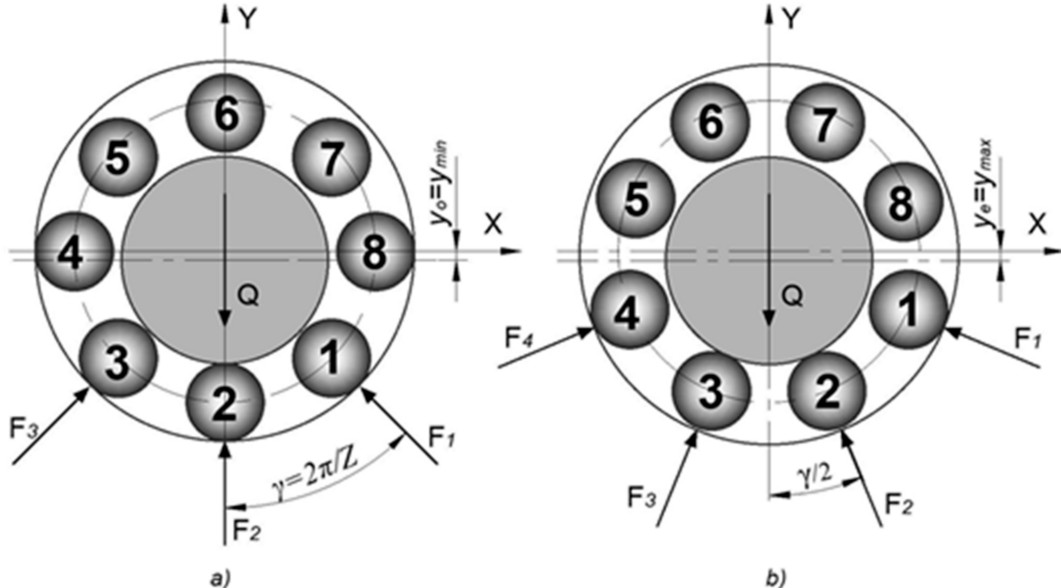

**Figure 1.** Boundary positions of inner rings support, (**a**) BSO position, (**b**) BSE position

BSO occurs when the center of one of the rolling elements coincides with the direction of the action of the external radial load. BSE occurs when the rolling elements are symmetrically positioned relative to the direction of the action of the external radial load. The minimum radial distance between the center of bearing rings appears in the BSO position, and in the BSE position the maximum radial distance appears.

The axis of the inner ring, in relation to the outer ring, will continuously oscillate between these two boundary positions of the support. These oscillations will occur in all bearings with an internal radial clearance, regardless of the number of rolling elements and the size of the external radial load.

### 2.2. Vibrations Caused by the Periodic Change of Rolling Bearing Stiffness

During the bearing operation, due to a contact load, in the contact between the rolling elements with the bearing rings, the elastic deformations occur. These deformations are non-linear and can be described according to *Hertz theory of the contact stresses and deformations* [1]:

$$\delta_i = \left(\frac{F_i}{K}\right)^{\frac{1}{n}},\tag{1}$$

where $\delta_i$ is contact deformation at the place of *i*-th rolling element, $F_i$ is load of *i*-th rolling element, $K$ is effective stiffness coefficient due to Hertz's contact effect, $n$ is the exponent that depends on bearing type ($n = 3/2$ for ball bearing and $n = 10/9$ for bearing with rollers).

With the change of position of the set of rolling elements, the load distribution in a bearing is changed, and with that intensity and the direction of contact forces and contact deformations between the rolling elements and rings also changes (Figure 2). All of this will cause a periodical change of the bearing stiffness [26,27]. The time-varying stiffness causes vibrations also in an ideally made bearing even in the absence of internal radial clearance. Moreover, the time-varying stiffness is considered to be a basic cause of the generation of the vibrations in the rolling bearings. The vibrations that occurred in this way are called variable compliance (VC) vibrations [28].

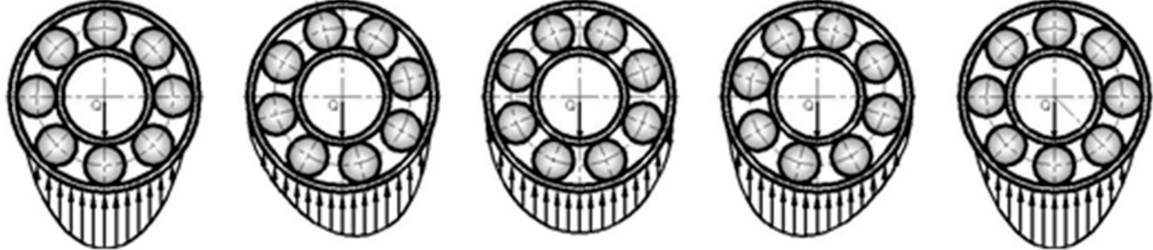

**Figure 2.** Periodic change of load distribution inside the rolling bearing.

## 3. The Equation of Rigid Rotor Vibration in Rolling Element Bearing

The equation for the rigid rotor vibrations in an ideal rolling bearing with the internal radial clearance was derived by Kryuchkov and Smirnov [17,18], in the form of an absolute sinusoid as:

$$y = \Delta \left| \sin \left( \frac{f_{bp}}{2} \cdot t \right) \right|, \tag{2}$$

where $\Delta$ is peak-to-peak (*pp*) amplitude, *t* is time, and $f_{bp}$ is ball passage frequency (BPF). The derivation of Equation (2) is described in details in the reference under ordinal number [20].

According to the Equation (2), the vibration frequency of a rigid rotor in an ideal rolling bearing with the internal radial clearance is equal to a BPF. BPF represents the speed with which the rolling elements pass over some fixed point on an outer bearing race. According to [1] BPF is calculated as:

$$f_{bp} = \frac{z}{2} \cdot \omega \cdot \left( 1 - \frac{d_b}{d_c} \cdot \cos \alpha \right), \tag{3}$$

where $\omega$ is the shaft frequency, $d_b$ is the diameter of rolling elements, $d_{c-}$ is the cage pitch diameter, $\alpha$ is the bearing contact angle.

The size of *pp*-amplitude ($\Delta$) can be calculated as the difference of extreme values of the rotor displacement in two boundary positions of support, shown in Figure 1, according to the equation:

$$\Delta = y_e - y_o, \tag{4}$$

where $y_e$ is displacement of the rotor in case *BSE* (Figure 1b), $y_o$ is displacement of the rotor in case *BSO* (Figure 1a).

The rotor displacement is determined by the size of the relative motion between the rings during the bearing operation. According to [19], a relative movement of bearing rings directly depends on the size of internal radial clearance, load distribution in bearing, construction of bearing, the level of contact deformations and errors in the geometry of the contact surfaces.

## 4. Theoretical Analysis of Rigid Rotor Vibrations Amplitude in Rolling Bearings

### 4.1. Level of pp-Amplitude in an Unloaded Rolling Bearing

In the case of an unloaded bearing or a bearing loaded with relatively low loads, the contact deformations are insufficient to annul the influence of the internal clearance and the inner ring will be supported on one or two rolling elements, as shown in Figure 1. For this case of support in literature [29], a term *support system 1-2* was adopted.

The rotor displacement in boundary positions of the support is possible to be calculated by the help of the Equations [20]:

$$y_e = \frac{e}{2 \cdot cos\gamma/2}, \tag{5}$$

$$y_o = \frac{e}{2},$$ (6)

where $e$ is internal radial clearance, $\gamma$ is angular distance between the rolling elements.

According to the Equation (4) the level of the *pp*-amplitude in a case of the absence of external radial load, it will be equal to:

$$\Delta_1 = y_e - y_o = \frac{e}{2} \cdot \left( \frac{1}{\cos(\gamma/2)} - 1 \right).$$ (7)

The detailed derivation of the above equation is shown in [20].

### 4.2. Impact of Load and Contact Deformations on Level of pp-Amplitude

Due to the load and the occurrence of contact deformations, additional displacement and approach of the inner ring of the bearing relative to the external will occur and the new rolling elements will come into contact with the bearing rings. The support of bearing will continue to happen between the two boundaries positions: on the even (BSE) and the odd (BSO) number of the rolling elements. Only, depending on the size of contact deformations, the support system will move from *the support system 1-2* to *the support system 2-3, 3-4, 4-5*, etc., as shown in Figure 3 [19,21].

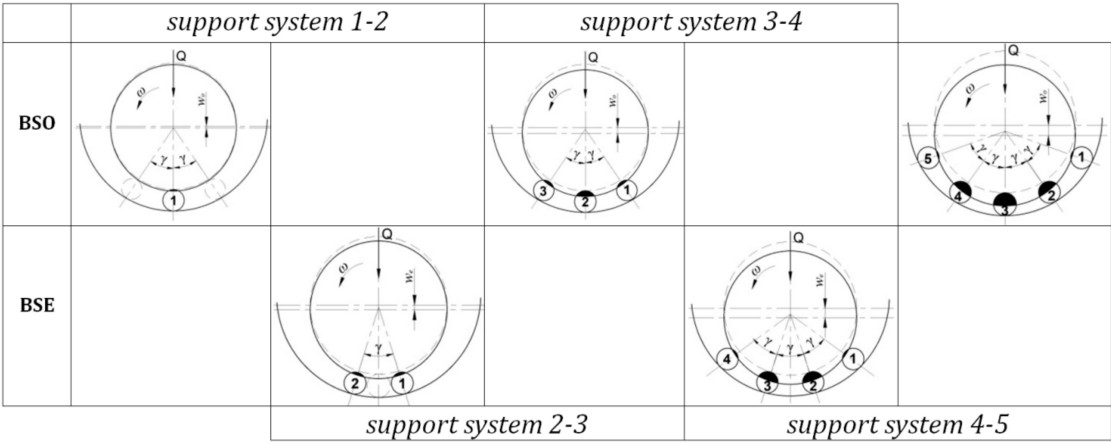

**Figure 3.** Support systems of rolling bearing.

The number of rolling elements that will come into contact with the bearing rings is directly dependent on two sizes, the inner radial clearance, and the external radial load, and can be easily determined by the procedure given in the literature [21,27].

The additional displacement of bearing rings will cause the further change of the *pp*-amplitude of rotor vibrations.

The total displacements of the rotor center will be equal to the sum of the displacements for unloaded bearing (Equations (5) and (6)) plus displacement due to the contact deformations of bearing elements (Figure 4). Hence, for the boundary positions of the supports on an even and an odd number of the rolling elements, the total displacement of the rotor center will be equal:

$$y_e = \frac{e}{2 \cdot \cos \gamma/2} + w_e,$$ (8)

$$y_o = \frac{e}{2} + w_o,$$ (9)

where $w_e$ and $w_o$ are rotor center displacements due to the contact deformations (bearing deflection) in the BSE and BSO cases of support system.

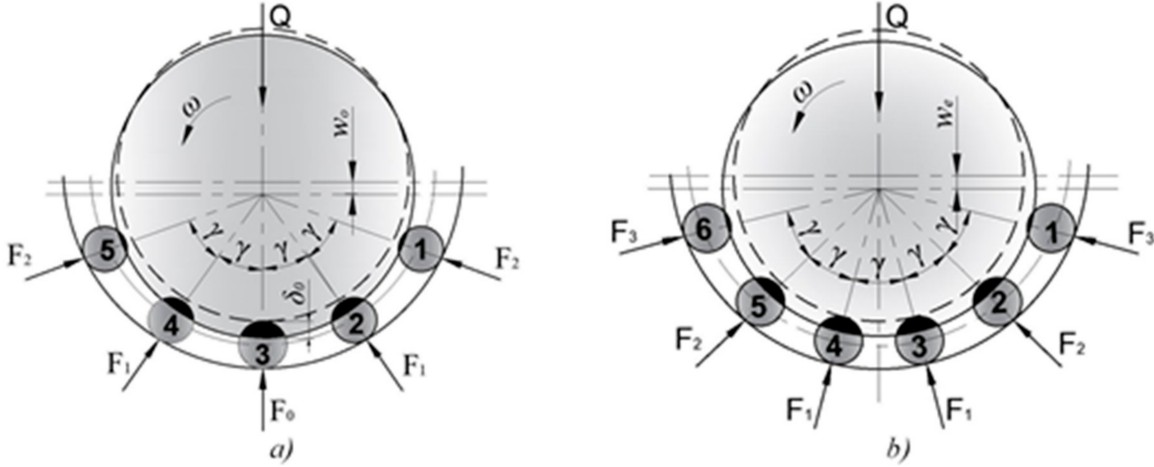

**Figure 4.** The influence of the external load and contact deformations on the relative displacement between the bearing rings, (**a**) BSO position of support, (**b**) BSE position of support.

According to the Equation (4), the amplitude of VC vibrations of the loaded rolling bearing can be calculated as a difference of the total center displacement for the boundary cases of the support, according to:

$$\Delta = y_e - y_o = \left( \frac{e}{2\cos(\gamma/2)} + w_e \right) - (e/2 + w_o), \tag{10}$$

$$\Delta = y_e - y_o = \frac{e}{2} \left( \frac{1}{\cos(\gamma/2)} - 1 \right) - (w_o - w_e) = \Delta_1 - \Delta_2, \tag{11}$$

where $\Delta_2 = w_o - w_e - pp-$ amplitude of the rotor vibrations due to the contact deformations of bearing elements. This amplitude is equal to a difference of the bearing deflection in two boundary support position: BSO ($w_o$) and BSE ($w_e$).

### 4.3. Radial Displacement of the Rotor Center Due to the Contact Deformations

In Figure 4 it is shown how due to contact deformations and relative displacement between the bearing rings, it comes to the annulment of the negative impact of the internal radial clearance and the entry of new rolling elements into the load transfer. The level of relative displacement and contact deformation is directly dependent on the size of the external radial load.

In [21], equations have been presented by which is possible to calculate the size of relative displacement of the bearing rings for BSO and BSE positions of support.

#### 4.3.1. BSO Position of Support (Figure 4a)

The dependence between the bearing deflection ($w_o$) and external radial load ($Q$) for the BSO position according to [22] can be obtained from the following equation (Figure 4a):

$$Q = K \cdot \left[ w_o^n + 2 \cdot \sum_{i=1}^{\frac{q-1}{2}} (w_o - a_{2i+1})^n \cdot \cos^{n+1}(i\gamma) \right], \tag{12}$$

where $i$ is index of rolling element, $K$ is effective coefficient of stiffness, $w_o$ is bearing deflection, $q$ is total number of rolling elements on which inner ring is supported, $a_{2i+1}$ is boundary deflection of bearing when an *i-th* rolling element enters into the contact with the bearing rings, $\gamma$ is angular spacing between rolling elements.

The bearing deflection that is necessary for entering of the *i-th* rolling element into the contact with the bearing rings in a BSO position can be calculated using the equation [21]:

$$a_{q,o} = \frac{e}{2} \cdot \left( \frac{1}{\cos \frac{q-1}{2} \gamma} - 1 \right) = t_{q,o} \cdot \frac{e}{2}, \tag{13}$$

where $t_{q,o}$ is the coefficient of the boundary deflection of bearing in the BSO position of inner ring support. The detailed derivation of the above equation and the value of coefficient $t_{q,o}$, can be found in the reference [19,21].

### 4.3.2. BSE position of support (Figure 4b)

The dependence between the bearing deflection ($w_e$) and external radial load ($Q$) for the BSE position according to [21] can be obtained from the following equation (Figure 4b):

$$Q = 2 \cdot K \cdot \sum_{i=1}^{\frac{q}{2}} (w_e - a_{2i,e})^n \cdot \cos^{n+1}(2i-1)\frac{\gamma}{2}, \tag{14}$$

where: $i$ is index of rolling element, $K$ is effective coefficient of stiffness, $w_e$ is bearing deflection, $q$ is total number of rolling elements on which inner bearing ring is supported, $a_{2i}$ is boundary deflection of bearing when an *i-th* rolling element enters into the contact with the bearing rings, $\gamma$ is angular spacing between rolling elements.

The necessary bearing deflection for entering of the *i-th* rolling element into the contact with bearing rings in the BSE position can be calculated as [21]:

$$a_{q,e} = \frac{e}{2} \cdot \left( \frac{1}{\cos \frac{q-1}{2} \gamma} - \frac{1}{\cos \frac{\gamma}{2}} \right) = t_{q,e} \cdot \frac{e}{2}, \tag{15}$$

where $t_{q,e}$ is the coefficient of the boundary deflection of bearing in the BSE position of inner ring support [19,21].

## 5. Parametric Analysis of the Amplitude of VC Vibration in Rolling Bearings

For the parametric analysis a radial single-row ball bearing 6206 with the internal radial clearance has been chosen. This bearing has nine balls altogether. According to [21], for this bearing, the maximum number of rolling elements that can emerge in the loaded zone is five. From the corresponding tables presented in [21], values of the coefficient of bearing boundary deflection $t_q$ were read. These values are presented in Table 1.

Table 1. The coefficient of bearing boundary deflection $t_q$ for ball bearing 6206.

|       | $q = 3$ | $q = 4$ | $q = 5$ |
|-------|---------|---------|---------|
| $t_q$ | 0.3054  | 0.9358  | 4.7588  |

### 5.1. Numerical Procedure

Equation (11) gives a general pattern for the calculation of the amplitude of the rigid rotor vibration in a rolling bearing. According to this equation, the total amplitude of the VC vibrations ($\Delta$) is equal to the difference of *pp*–amplitude of an unloaded bearing ($\Delta_1$) and *pp*-amplitude due to the contact deformations ($\Delta_2$), i.e.:

$$\Delta = \Delta_1 - \Delta_2. \tag{16}$$

The size of *pp*-amplitude in the case of an unloaded bearing can be obtained with the Equation (7). On the other side, the *pp*-amplitude due to the contact deformations is equal to the difference of the bearing deflection in the BSO and BSE positions, according to the equation:

$$\Delta_2 = w_o - w_e. \tag{17}$$

The values of the bearing deflection can be obtained by solving Equation (12) for the case of the support on an odd number of rolling elements, and/or Equation (14) for the case of the support on an even number rolling elements. Given equations are non-linear and can only be calculated numerically. The bisection method was used for equation solving. For that purpose, computer software that runs in the MATLAB programming environment has been developed [29].

### 5.2. Unloaded Rolling Bearing

In the case of an unloaded rolling bearing, the amplitude of the rotor vibration depends on the size of the clearance and the angular spacing of rolling elements (Equation (7)). As the angular spacing was directly determined by the number of rolling elements in a bearing, it can be said that this amplitude depends on the total number of rolling elements and the size of the internal radial clearance and that other characteristic of bearing do not influence on its size. This means that an unloaded radial bearing with the same number of rolling elements and the size of clearance will have the same level of vibrations regardless of the type and dimensions of its constituent parts.

The diagram in Figure 5 gives a three-dimensional dependence of the vibration amplitude on the number of rolling elements and the internal clearance in an unloaded rolling bearing.

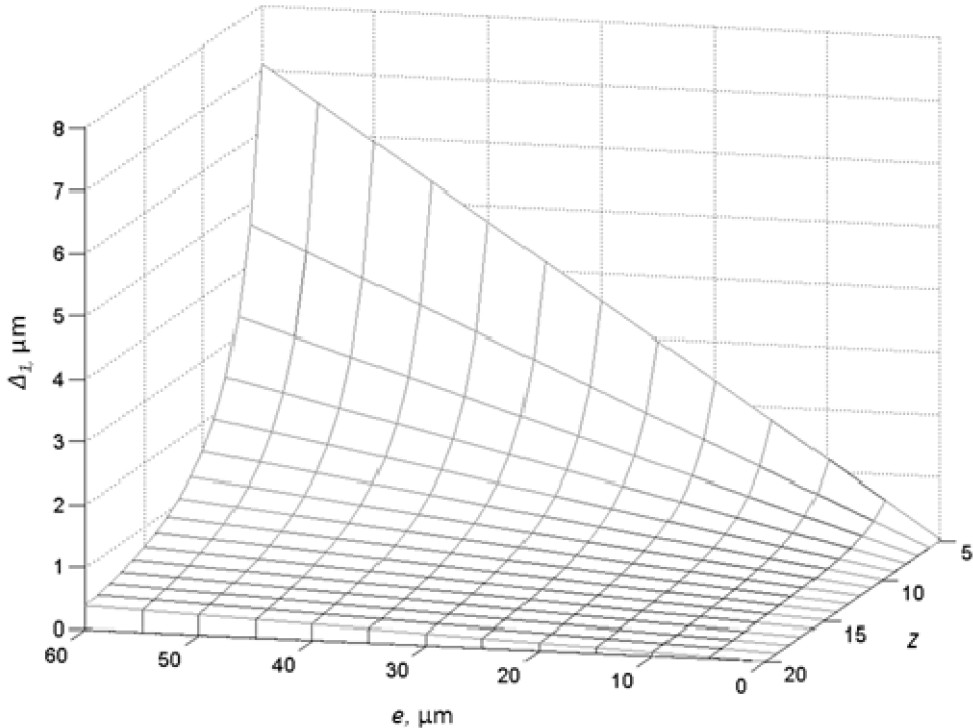

**Figure 5.** The vibrations amplitude of the unloaded rolling bearing depending on the size of the internal radial clearance and the total number of rolling elements.

From the analysis of the diagram by the increase of the internal clearance, the size of vibration amplitude linearly increases, and this increment is more pronounced when the number of rolling elements in a bearing is smaller. This means that more rotation precision and lower level of rotor

vibrations are accomplished by the reduction of the angle between the rolling elements i.e., with the increase of the total number of rolling elements in a bearing. However, for construction reasons, the increase in the number of rolling elements is impossible over a certain limit. That means that the rotor vibrations in an unloaded rolling bearing are unavoidable and that this is a structural characteristic of a rolling bearing, which is impossible to avoid.

### 5.3. Loaded Rolling Bearing the Influence of Internal Radial Clearance and External Load

Figures 6–9 present the three-dimensional diagrams of the bearing deflection ($w_o$) and ($w_e$) in BSO and BSE positions, as well as the sizes of the vibration amplitude due to the contact deformations ($\Delta_2$) and the total amplitudes of VC vibrations ($\Delta$), depending on the size of the internal radial clearance and external radial load, for the 6206 ball bearing. The value of the internal radial clearance was varied in the range from 0 to 60 μm on the diagrams. That is nearly equal to the maximal recommended value of internal radial clearance for the 6206 ball bearing [30]. The value of the external radial load was varied up to a static load rating of the bearing 6206, which is $Q$ = 11200 N.

The diagrams of the dependence of relative displacement of bearing rings ($w_o$) and ($w_e$) due to the contact deformations are given in Figures 6 and 7. The contact deformations of bearing elements increase together with the increase of the level of external radial load, but also with the increase of internal radial clearance size. However, the increase of the contact deformations is fairly more pronounced for the increase of the external radial load. The fact that the zones of various colors are almost parallel with the axis of internal radial clearance clearly indicates this. Also, it can be observed that there is a somewhat higher level of bearing deflection, at support on an odd ($w_o$) in relation to the support on an even ($w_e$) number of rolling elements. As the amplitude of the vibrations due to contact deformation ($\Delta_2$) is equal to the difference of the bearing deflection in the BSO and BSE positions (Equation (17)), the level of this amplitude has a positive value for all the combination of the external radial load and internal radial clearance of bearing. This is also shown in the diagram shown in Figure 8.

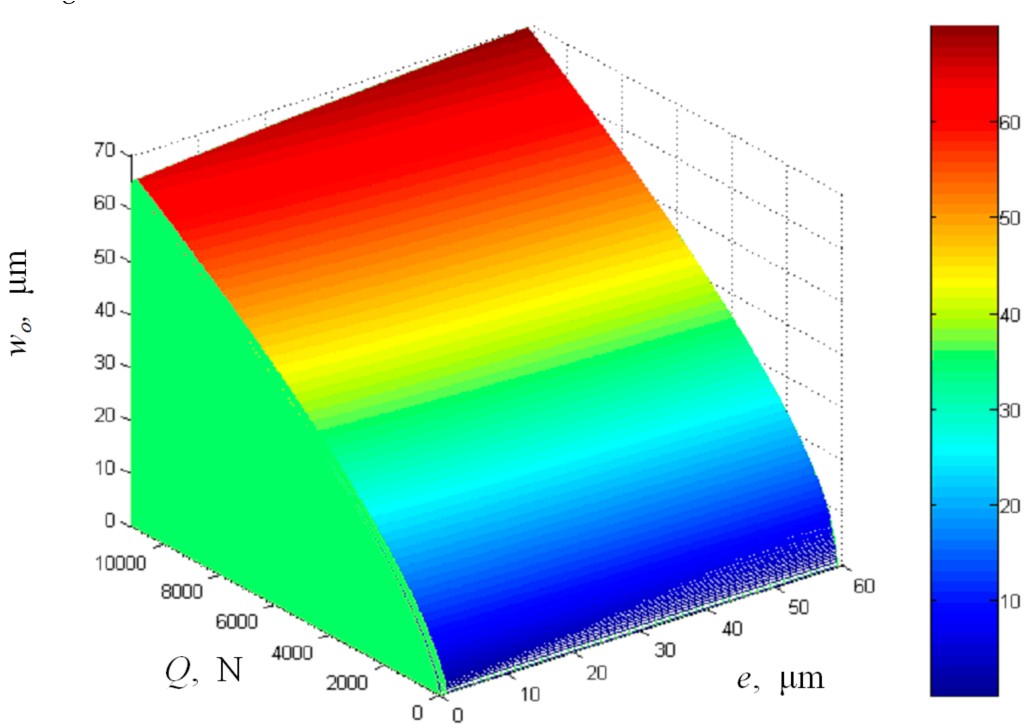

**Figure 6.** Three-dimensional dependence of bearing deflection ($w_o$) on the size of internal radial clearance ($e$) and level of external load ($Q$), for the BSO position of 6206-single-row ball bearing.

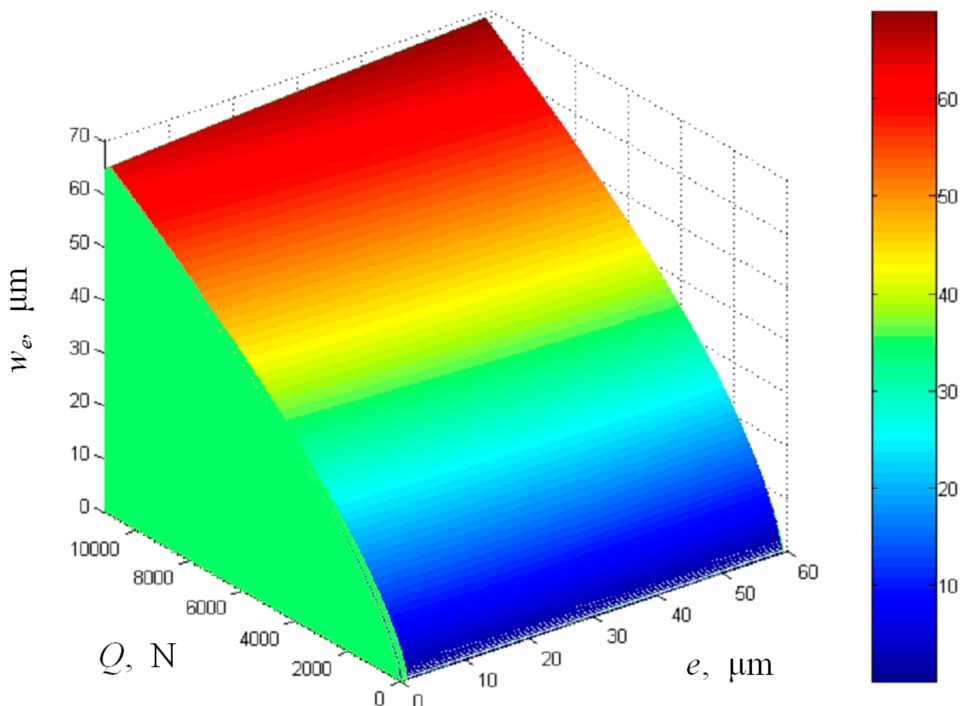

**Figure 7.** Three-dimensional dependence of bearing deflection ($w_e$) on the size of internal radial clearance ($e$) and level of external load ($Q$), for the BSE position of 6206-single-row ball bearing.

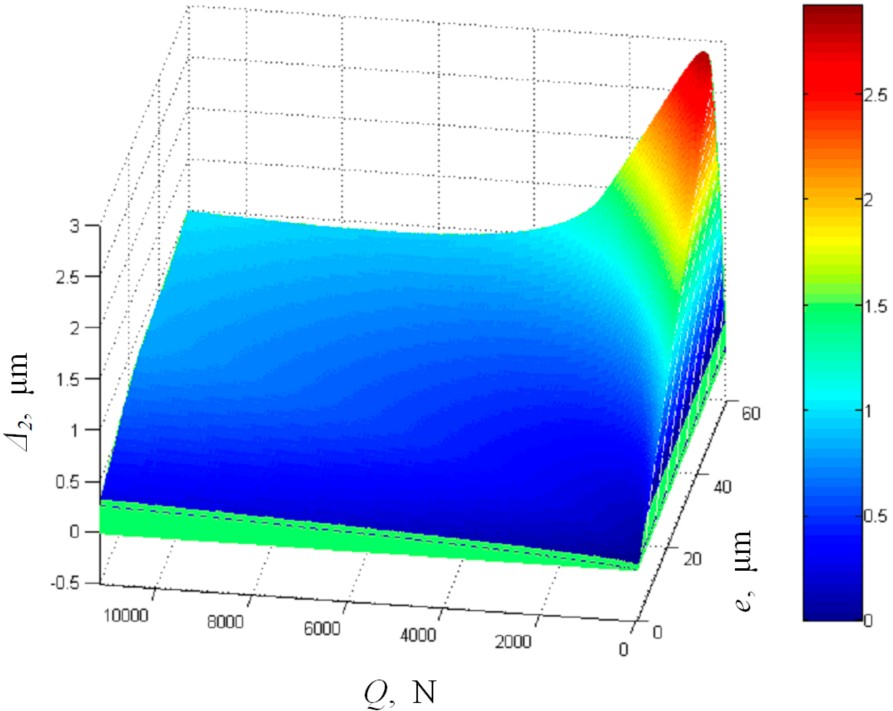

**Figure 8.** Three-dimensional dependence of vibration amplitude due to contact deformations ($\Delta_2$) on the size of internal radial clearance ($e$) and level of external load ($Q$) for the 6206-single-row ball bearing.

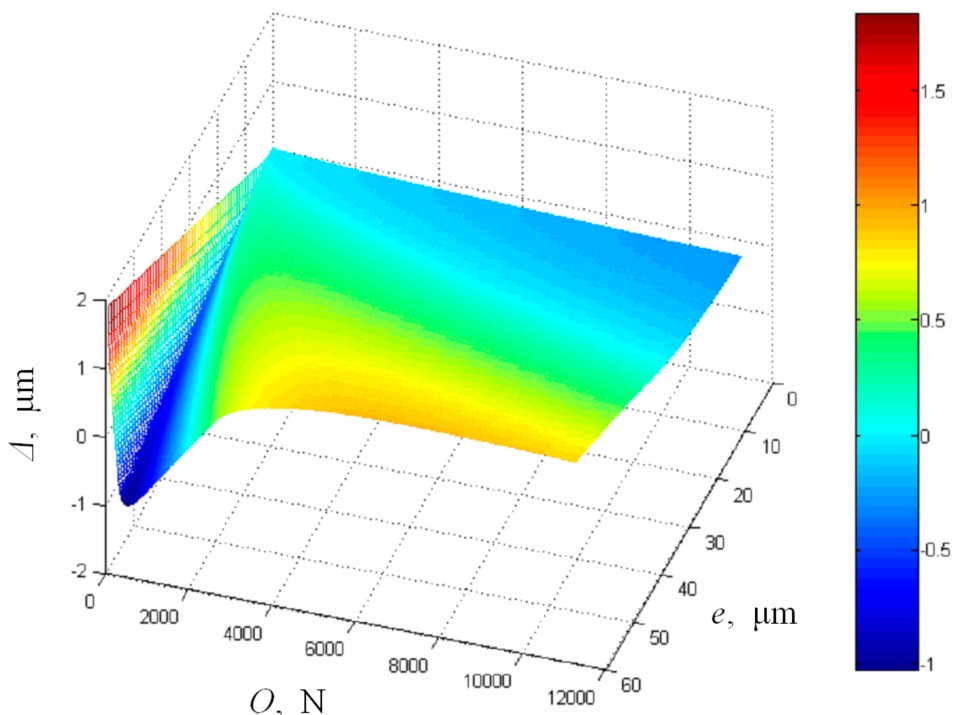

**Figure 9.** Three-dimensional dependence of the total amplitude of VC vibration ($\Delta$) on the size of internal radial clearance ($e$) and level of external load ($Q$) for the 6206-single-row ball bearing.

In Figure 8 the change of the *pp*-amplitude due to the contact deformations ($\Delta_2$) has been presented. In relation to the level of external radial load, we can discern two characteristic areas in this diagram. The first area is characterized by a high gradient of vibration amplitude due to the contact deformations ($\Delta_2$). In this area, the support of the inner ring is carried out by *support systems 1-2,* or possibly by *support systems 2-3* (see Section 4.2). Because of the low intensity, the external radial load is not enough for the entering of a bigger number of rolling elements into contact with the bearing rings. The smaller number of rolling elements that transfer the external load, results in a less convenient load distribution into the bearing and in a sharper gradient of contact deformations. Hence, the steeper shape of the diagram of *pp*- amplitude $\Delta_2$ in the direction of the external radial load in this area. On the other side, and increasing the clearance will also cause an increase in the level of the amplitude $\Delta_2$ in this area, except for *support system 1-2.*

In *support systems 1-2*, the size of the clearance does not affect the vibration amplitude $\Delta_2$. The vibration amplitude due to the contact deformation stays constant, regardless of the increase of internal radial clearance in *support systems 1-2*. This is also shown by the contour lines that are parallel to the axis that marks the value of the internal radial clearance ($e$). This area is characterized by a relatively low intensity of the external radial load.

The increase of external radial load significantly increases the level of the vibration amplitude due to the contact deformations. In the case of a boundary load that will cause the boundary deflection of bearing, the bearing support system will move from *support systems 1-2* to *support systems 2-3*. This phenomenon is explained in Section 4.2 and more comprehensively in the papers under the ordinal number [21,22]. Further increase in load will result in a negative gradient of the amplitude $\Delta_2$.

In the second area, the high level of the external load results in a bigger number of rolling elements into contact with bearing rings. Hence, the load distribution between the rolling elements is fairly more uniform in this area. This results in the reduction of the vibration amplitude due to the contact deformations ($\Delta_2$). The increase of the external radial load will reduce the amplitude level ($\Delta_2$) until the point when an inner ring begins to support on a maximal possible number of active rolling elements. After this point, the amplitude will increase again.

The diagram of the total amplitude of VC vibrations in a rolling bearing has been presented in Figure 9. Unlike the vibration amplitude due to the contact deformations, the total amplitude of the VC vibrations can have both positive and negative values. The total amplitude has most values in the areas of the lower intensity of external radial load and the bigger values of internal radial clearance. These are the areas where, because of a low level of contact deformations, the load transfer is mostly based on *support systems 1-2*. The biggest influence on the rotor vibrations in this area has the size of the internal radial clearance. For the smaller values of clearance, the level of the total vibration amplitude is also smaller.

The external radial load has a favorable influence on the rotor vibrations only in the areas where the total vibration amplitude is positive. Namely, in the areas where emerges an unfavorable load distribution, i.e., in the areas where inner ring support on a smaller number of rolling elements, the level of the contact deformations can be very high so that the vibration amplitude due to contact deformations is bigger than amplitude $\Delta_1$. In these areas, the value of the total amplitude is negative. In the areas where the total amplitude has a negative value, a bigger influence on rotor vibrations has contact deformations than the size of internal radial clearance. In the fields where the total amplitude is positive, a bigger influence on the rotor vibrations has the clearance in a bearing than the size of the external radial load.

### 5.4. Loaded Rolling Bearing the Influence of the Total Number of Rolling Elements and External Load

The diagrams in Figures 10–13. show the influence of the external radial load and the total number of rolling elements on the basic parameters that define the VC vibrations of the rolling bearing (bearing deflection ($w_o$) and ($w_e$), *pp*-amplitude due to contact deformations ($\Delta_2$) and total *pp*-amplitude ($\Delta$)). In order to exclude the influence of the internal bearing construction, the analysis was made with respect to the ratio of the external radial load and the effective bearing stiffness coefficient ($Q/K$). The analysis was made for bearings with an internal radial clearance of $e = 15$ μm.

It is clear that the increase of the external radial load causes an increase of the contact deformations in bearing and, consequently, an increase in the bearing deflection. However, for bearings with a larger number of rolling elements, this increase is much milder. The diagrams in Figures 10 and 11 confirm that an increase in the total number of rolling elements in the bearing significantly influences the reduction of bearing deflection.

In Figure 12, the change in *pp*-amplitude due to contact deformations ($\Delta_2$) is shown. In relation to the total number of rolling elements in the bearing, the diagram in the figure can be divided into two characteristic areas. The first area refers to bearings with a small total number of rolling elements ($z \leq 6$). In this area, with increasing external radial load, the level of *pp*-amplitude ($\Delta_2$) increases significantly. For these bearings, the maximum number of roller bearings that can be found in the load zone is $z_s = 3$, that is, the external load is transferred according to *the support system 2-3*. In Section 5.3, it is shown that in *the support system 2-3*, very high values of the *pp*-amplitude ($\Delta_2$) occur and the total VC vibrations are more influenced by the contact deformations. In the second area, due to the larger number of rolling elements in the bearing, the contact deformations are smaller and the level of *pp*-amplitude $\Delta_2$ decreases.

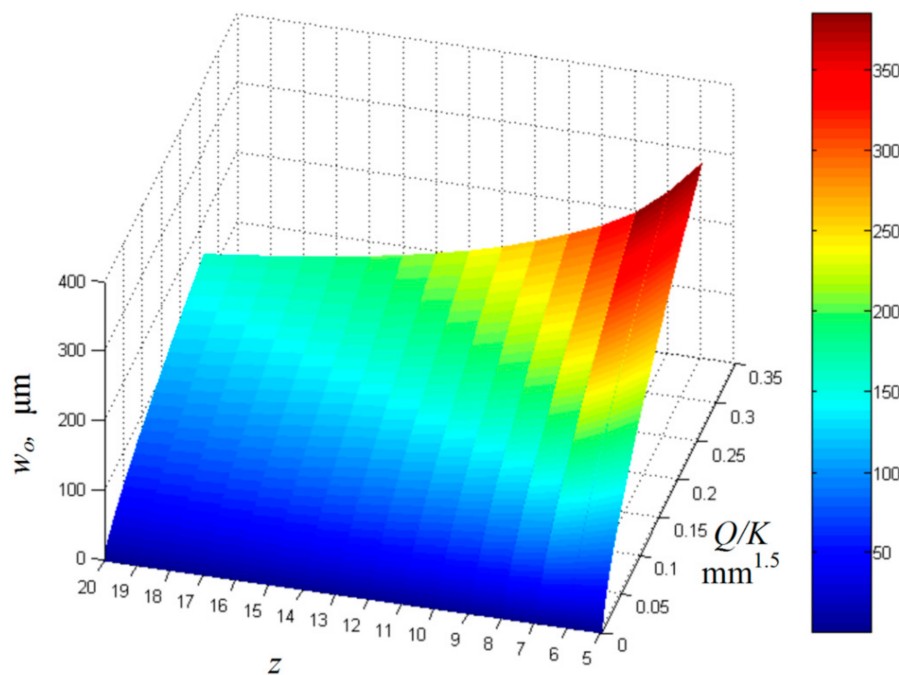

**Figure 10.** Three-dimensional dependence of bearing deflection ($w_o$) on the total number of rolling elements ($z$) and level of external load ($Q$), for the BSO position of rolling bearing.

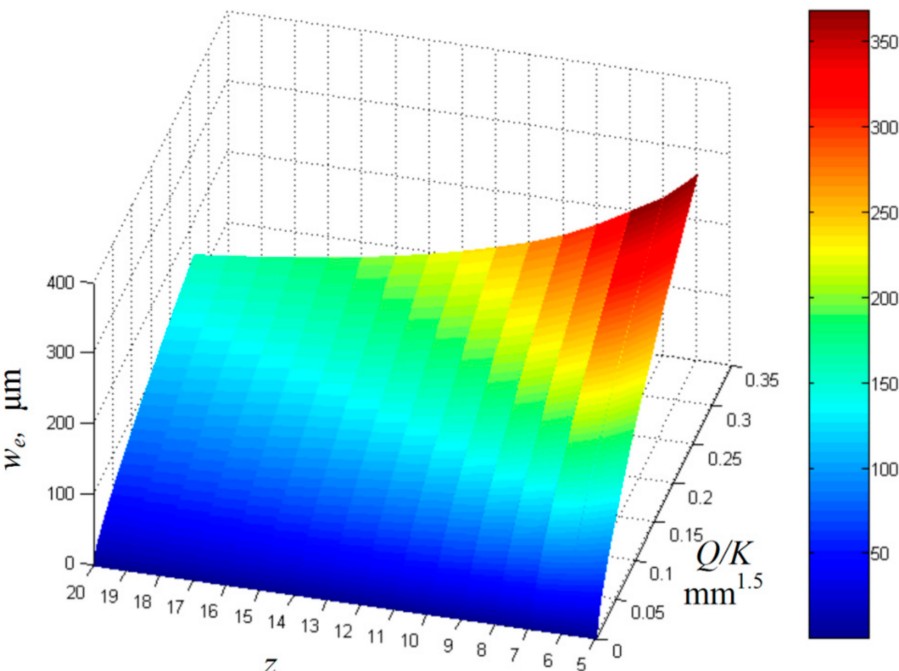

**Figure 11.** Three-dimensional dependence of bearing deflection ($w_e$) on the total number of rolling elements ($z$) and level of external load ($Q$), for the BSE position of rolling bearing.

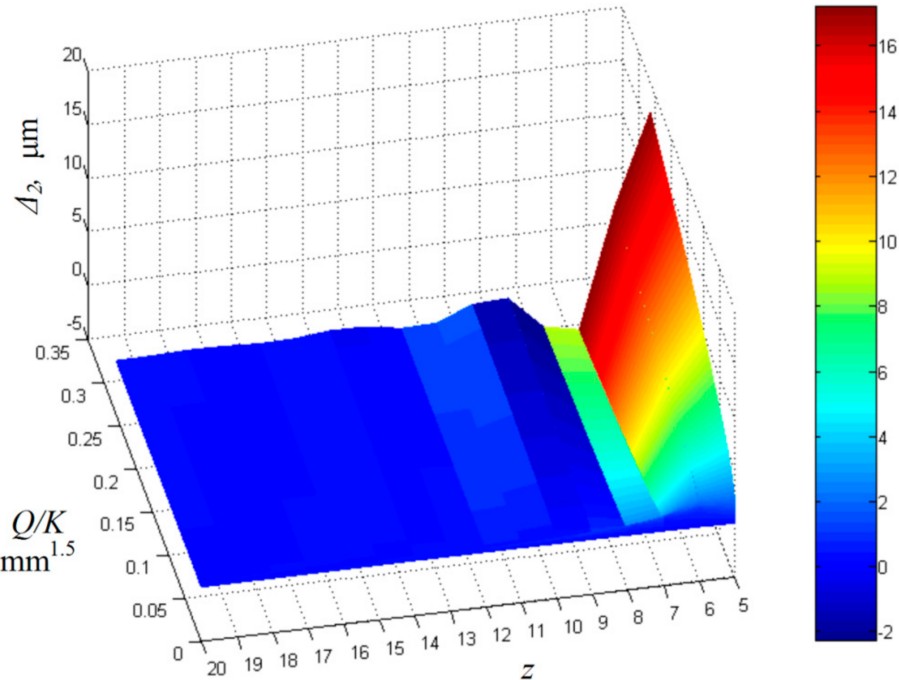

**Figure 12.** Three-dimensional dependence of vibration amplitude due to contact deformations ($\Delta_2$) on the total number of rolling elements ($z$) and level of external load ($Q$).

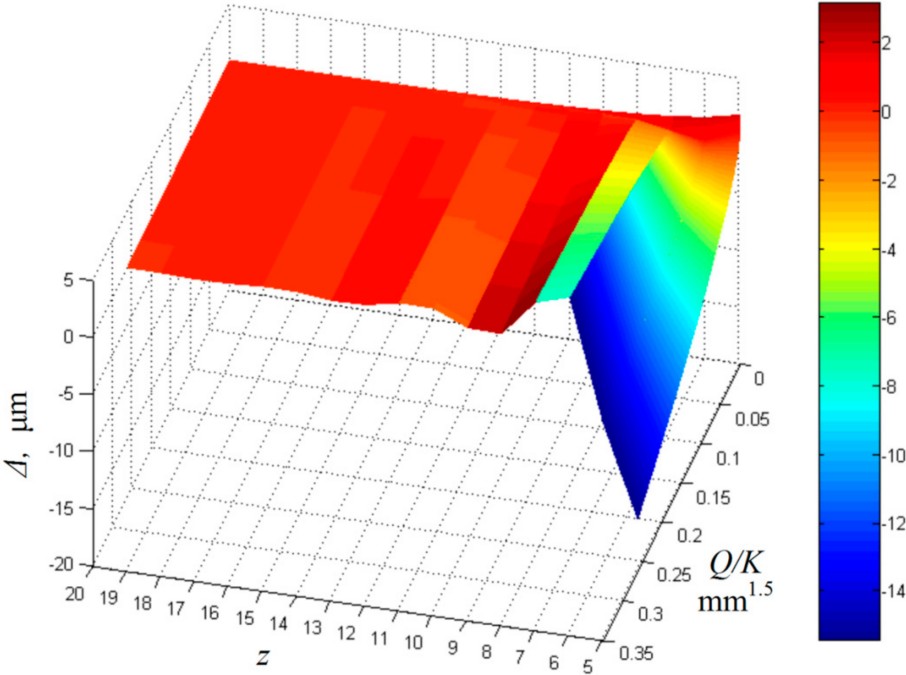

**Figure 13.** Three-dimensional dependence of the total amplitude of VC vibration ($\Delta$) on the total number of rolling elements ($z$) and the level of external load ($Q$).

In Figure 13, the change of the total *pp*-amplitude of VC vibrations ($\Delta$) is shown. Three characteristic areas can be distinguished in this diagram. The first two areas are characterized by quite high values of the total *pp*-amplitude. In the first area, the *pp*-amplitude takes values up to −15 μm and in the second area up to 8 μm. The first area refers to bearings with a total number of rolling elements $z \leq 6$. Due to the small number of rolling elements the contact deformations are large in this area. Therefore, in the first region, the influence of contact deformations on VC vibrations is predominant, and the

total *pp*-amplitude (Δ) is negative. The second area refers to bearings with a total number of rolling elements from 7 to 9. Due to the larger number of rolling elements, a significant decrease in amplitude ($\Delta_2$) occurs in this region, so high values of total *pp*-amplitude are a consequence of high values of amplitude $\Delta_1$. The values of total *pp*-amplitude in this region are positive. The third area on the diagram is characterized by a relatively low value of total *pp*-amplitude. The value of *pp*-amplitude in this region decreases with the increase of the total number of rolling elements in the bearing. This confirms the conclusion that the greater total number of rolling elements in the bearing favorably affects the dynamic characteristics of the bearing.

## 6. Discussion

Shown diagrams demonstrate that there is a narrow link between the influence of internal radial clearance and the external radial load on the basic parameters that define the vibration response of a rolling bearing.

Depending on the combination of the level of external radial load and the size of internal radial clearance, the level of a bearing deflection is changed, as well as the number of rolling elements participating in the distribution of load in bearing. This also changes the vibration response of rolling bearing. This gives an opportunity that analysis of VC vibrations of rolling bearing can be also used as an important diagnostic parameter for the estimation of the internal radial clearance size and the uniformity of the internal load distribution, as well as the number of rolling elements which participate in the transfer of external load.

According to Equation (16), the vibrations that emerged due to the influence of internal radial clearance can be annulled by the vibrations that emerged due to the contact deformations. This is very important because by the appropriate combination of external radial load and internal radial clearance is possible to project the level of the VC vibrations of rolling bearing and reduce these vibrations to a minimum value.

## 7. Conclusions

Based on the results presented above, the following can be concluded:

- In this paper, a new simplified mathematical model for the calculation of amplitude of varying compliance vibrations is developed.
- The amplitude of the varying compliance vibrations in a rolling bearing is equal to the difference between vibration amplitude in an unloaded rolling bearing and the amplitude emerged due to the contact deformations.
- The vibration amplitude of an unloaded rolling bearing is determined by the size of the internal radial clearance and with the total number of rolling elements in a bearing.
- The vibration amplitude due to the contact deformations is equal to the difference of bearing deflection in two boundary positions of the support on an odd and an even number of rolling elements.
- The parametric analysis showed that the size of the internal radial clearance and the level of the external radial load have the biggest influence on the level of varying compliance vibrations.
- Higher values of the external radial load and the lower values of the internal radial clearance affect the VC vibrations level favorably.
- The greater total number of rolling elements in the bearing favorably affects the dynamic characteristics of the bearing
- The correct choice of the combination of the internal radial clearance and the external radial load can theoretically reduce the VC vibrations level to zero.

**Funding:** This research received no external funding.

**Conflicts of Interest:** The authors declare no conflict of interest.

## Nomenclature

| | |
|---|---|
| *VC* | varying compliance |
| *BSO* | boundary case of inner ring support on an odd number of rolling elements |
| *BSE* | boundary case of inner ring support on an even number of rolling elements |
| $\Delta$ | contact deformation |
| *F* | rolling element load |
| *K* | effective coefficient of stiffness due to Hertz's contact effect |
| *n* | exponent |
| *y* | rotor displacement |
| $\Delta$ | peak-to-peak (*pp*) amplitude of VC vibration |
| $\Delta_1$ | peak-to-peak (*pp*) vibration amplitude of unloaded bearing |
| $\Delta_2$ | peak-to-peak (*pp*) vibration amplitude due to the contact deformations |
| *BPF* | ball passage frequency |
| $f_{bp}$ | ball passage frequency |
| *t* | time |
| *z* | total number of rolling elements |
| $\omega$ | shaft frequency |
| $\alpha$ | bearing contact angle |
| $d_c$ | cage pitch diameter |
| $d_b$ | diameter of rolling elements |
| *w* | bearing deflection |
| $\gamma$ | angle between rolling elements |
| *e* | internal radial clearance |
| *q* | number of active rolling elements |
| $a_q$ | boundary deflection of bearing |
| $t_q$ | coefficient of boundary deflection of bearing |
| *Q* | total bearing load |

## Indexes

| | |
|---|---|
| *i* | index of rolling element |
| *e* | boundary case of support onto an even number of rolling elements |
| *o* | boundary case of support onto an odd number of rolling elements |

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
