# Peer review of "A Simplified Mathematical Model for the Analysis of Varying Compliance Vibrations of a Rolling Bearing"

_applsci, doi:10.3390/app10020670_

Round 1
Reviewer 1 Report
The author proposed a simple model to analyze the compliance vibration of rolling bearing. suggestions and comments are as follows:
1. It would be better to improve the writing of the paper, otherwise, it is a bit difficult to read and understand it.
2. Since the model is a deep simplified model, which means the model is not able to take consider many conditions of the practical bearing. The author also illustrates the result of the model is opposite to a general conclusion of the varying compliance vibration of rolling bearing. It would be better to check the model again.
3. The introduction part is not well- structured, it is hard to get the main idea of the research. The references from 5-15 can be categorized into groups, then get the conclusive remarks.
4. Please emphasize the characteristics of the references from line 65 to 98, meanwhile, point out their shortcomings on vibration response prediction.
5. Please explain the nomenclature in equations, for example, γ in eqs5. The contour diagram in fig. 6-9 is unnecessary to list separately, they can be plotted in the projection of the original diagram.
6. Since the influence of the total number of rolling elements on vibration amplitude under the unloaded condition has been considered in fig.5. Please supplement the vibration caused by the varying number of rolling elements under the load varying conditions in the following part. Otherwise, the simulation results are insubstantial.
Author Response
Dear editors and reviewers:
I am thankful to you very much for the constructive comments and suggestions for my manuscript entitled ‚‚A simplified mathematical model for the analysis of varying compliance vibrations of a rolling bearing".
The manuscript has been revised carefully based on the comments from the reviewer and I have marked the changes by using the ‚‚Track changes“ in the revised manuscript.
I deeply appreciate all of your help and suggestions for my manuscript. Please don’t hesitate to make new suggestions if you have any observations.
1. Moderate English changes required. It would be better to improve the writing of the paper, otherwise, it is a bit difficult to read and understand it.
Thanks for your suggestion. The complete manuscript has been revised. The grammar issues were checked carefully and the language part of this manuscript was revised. The manuscript has been checked by my two colleagues who speak English very well.
2. Since the model is a deep simplified model, which means the model is not able to take consider many conditions of the practical bearing. The author also illustrates the result of the model is opposite to a general conclusion of the varying compliance vibration of rolling bearing. It would be better to check the model again.
The statement "the result of the model is the opposite to the general conclusion of the varying compliance vibration of the rolling bearing" was incorrectly stated in the manuscript, because of the poor English translation. That is not a „general conclusion", as is written. The statement that this is a „general conclusion“ is corrected in the abstract and manuscript.
It is true that a large number of authors are of the opinion that the varying compliance vibration of rolling bearings cannot be avoided. However, there are studies that suggest the opposite. Such are e.g. the papers presented in references 17 and 18. My manuscript is based on this researches.
I can assure you that the proposed model is correct. It has been checked repeatedly and It applies only to study of the varying compliance vibration. Observation of an ideal bearing is sufficient to study them.
3. The introduction part is not well- structured, it is hard to get the main idea of the research. The references from 5-15 can be categorized into groups, then get the conclusive remarks. Please emphasize the characteristics of the references from line 65 to 98, meanwhile, point out their shortcomings on vibration response prediction.
Thanks for your suggestion. The complete introduction has been revised. References 5-15 are described as suggested by the reviewer.
The references from line 65 to 98 were used as the background for the research described in this paper. Some of them do not deal only with the vibration of the bearings.
4. Please explain the nomenclature in equations, for example, γ in eqs5.
I appreciate your suggestion. The nomenclature for all equation is added in the manuscript. For improving the readability of the manuscript, the nomenclature was added after the first page too.
5. The contour diagram in fig. 6-9 is unnecessary to list separately, they can be plotted in the projection of the original diagram.
I appreciate your suggestion. The contour diagrams are omitted from Figures 6-9.
6. Since the influence of the total number of rolling elements on vibration amplitude under the unloaded condition has been considered in fig.5. Please supplement the vibration caused by the varying number of rolling elements under the load varying conditions in the following part. Otherwise, the simulation results are insubstantial.
I appreciate your suggestion. Analysis of the vibration caused by the varying number of rolling elements under the load varying conditions was added in chapter 5.4.
Reviewer 2 Report
Memorandum
Subject: Review, December 27, 2019
Journal of Applied Sciences
Title: A simplified mathematical model for the analysis of varying compliance vibrations of a rolling bearing
Radoslav Tomović
University of Montenegro, Mechanical Engineering Faculty, 81000 Podgorica, Republic of
Montenegro;
Correspondence: radoslav@ucg.ac.me
Comments:
The authors should add a nomenclature to identify the parameters and abbreviations used throughout the paper. Matlab software should be referenced. The discussion of the results should be tied up to the data obtained, in its present form, it sounds more like a conclusion. The conclusion is rather long, it can be shortened and improved by stating what was achieved and stating what may have impacted the outcome of the results if any exist. Additionally, the author should consider bullet type statements citing the accomplishments derived from this research.
Overall the paper is in good format, upon addressing the above, it can be released for publication.
Author Response
Dear editors and reviewers:
I am thankful to you very much for the constructive comments and suggestions for my manuscript entitled ‚‚A simplified mathematical model for the analysis of varying compliance vibrations of a rolling bearing".
The manuscript has been revised carefully based on the comments from the reviewer and I have marked the changes by using the ‚‚Track changes“ in the revised manuscript.
I deeply appreciate all of your help and suggestions for my manuscript. Please don’t hesitate to make new suggestions if you have any observations.
1. The authors should add a nomenclature to identify the parameters and abbreviations used throughout the paper.
I appreciate your suggestion. The nomenclature was added after the first page.
2. Matlab software should be referenced.
The Matlab software was referenced in the reference list under number 29.
3. The discussion of the results should be tied up to the data obtained, in its present form, it sounds more like a conclusion.
In the phase of writing the discussion, the instructions for preparing the manuscript which is available on the Aplied Science journal website were used. Here is clearly advised that in a results discussion: "Authors should discuss the results and how they can be interpreted in perspective of previous studies and of the working hypotheses. The findings and their implications should be discussed in the broadest context possible and limitations of the work highlighted. Future research directions may also be mentioned. This section may be combined with Results."
The analysis of the results was made in the previous chapter. This is in accordance with the guidelines for the preparation of the manuscript, which states that the results chapter should state "Provide a concise and precise description of the experimental results, their interpretation as well as the experimental conclusions that can be drawn."
4. The conclusion is rather long, it can be shortened and improved by stating what was achieved and stating what may have impacted the outcome of the results if any exist. Additionally, the author should consider bullet type statements citing the accomplishments derived from this research.
Thanks for your suggestion. The conclusion was revised as it was suggested by the reviewer. The conclusion text is based on bullet type statements.
Round 2
Reviewer 1 Report
I appreciate the considerable effort put by the authors in revising the manuscript. Their responses to my comments are largely satisfactory. Hence, I recommend that the revised paper may now be accepted for publication in Applied Sciences.